# Association between heavy metal exposure and bacterial vaginosis: A cross-sectional study

Yu-Xue Feng[1,2], Ming-Zhi Tan[1,2], Hui-Han Qiu[1,2], Jie-Rong Chen[1,2], Si-Zhe Wang[1,3], Ze-Min Huang[1,4], Xu-Guang Guo[1,4,5,6]*

1 Department of Clinical Laboratory Medicine, Guangdong Provincial Key Laboratory of Major Obstetric Diseases, Guangdong Provincial Clinical Research Center for Obstetrics and Gynecology, The Third Affiliated Hospital, Guangzhou Medical University, Guangzhou, China, 2 Department of Clinical Medicine, The First School of Clinical Medicine, Guangzhou Medical University, Guangzhou, China, 3 Department of Clinical Medicine, The Nanshan College of Guangzhou Medical University, Guangzhou, China, 4 Department of Clinical Medicine, The Third School of Clinical Medicine, Guangzhou Medical University, Guangzhou, China, 5 Guangzhou Key Laboratory for Clinical Rapid Diagnosis and Early Warning of Infectious Diseases, King Med School of Laboratory Medicine, Guangzhou Medical University, Guangzhou, China, 6 Department of Obstetrics and Gynecology, Center for Reproductive Medicine, Guangdong Provincial Key Laboratory of Major Obstetric Diseases, Guangdong Provincial Clinical Research Center for Obstetrics and Gynecology, Guangdong-Hong Kong-Macao Greater Bay Area Higher Education Joint Laboratory of Maternal-Fetal Medicine, The Third Affiliated Hospital, Guangzhou Medical University, Guangzhou, China

* gysygxg@gmail.com

**Data Availability Statement:** The data are held in the public repository: NHANES database, which the

## Abstract

Bacterial vaginosis (BV) is a prevalent cause of vaginal symptoms in women of reproductive age. With the widespread of heavy metal pollutants and their harmful function on women's immune and hormonal systems, it is necessary to explore the association between heavy metal exposure and BV. This study investigates the potential relationship between serum heavy metals and bacterial vaginosis in a cohort of American women. The present study employed a cross-sectional analysis of 2,493 women participating in the 2001–2004 National Health and Nutrition Examination Survey (NHANES). Multivariable logistic regression models were utilized in the study to assess the correlation between these variables. A stratified analysis was performed to investigate the relationship among different population groups further, and smooth curve fittings were conducted to intuitively evaluate the correlation. According to the current cross-sectional study results, a significant correlation was identified between the high levels of lead and cadmium in the serum and the likelihood of developing bacterial vaginosis. We found that serum lead (OR = 1.35, 95% CI: 1.06–1.72, p = 0.016) and serum cadmium (OR = 1.41, 95% CI: 1.01–1.98, p = 0.047) increased the risk of bacterial vaginosis by 35% and 41%, respectively, in the highest level group in comparison to the lowest level group in the fully adjusted model. Furthermore, the research discovered no statistically significant association between the levels of total mercury in the serum and a heightened susceptibility to bacterial vaginosis (OR = 0.96, 95% CI: 0.75–1.23, p = 0.763). Results of our study indicated an inverse association between serum heavy metals and bacterial vaginosis risk, including lead and cadmium. Reducing exposure to heavy metals could be vital to preventing and managing bacterial vaginosis.

URL is https://www.cdc.gov/nchs/nhanes/about_nhanes.htm.

**Funding:** The author(s) received no specific funding for this work.

**Competing interests:** The authors have declared that no competing interests exist.

## 1. Introduction

Bacterial vaginosis (BV), which is a frequent source of vaginal issues in women of reproductive age, can be either symptomatic or asymptomatic [1]. Roughly half of female individuals experience symptoms such as vaginal malodor, discharge, itching, and an increase in vaginal pH [2]. A systematic review conducted in 2013 revealed that the prevalence of BV exhibits considerable variation between and within countries worldwide: African-American (51%), Hispanic (32%), and white (23%) [3]. A study based on the NHANES database showed that the prevalence of bacterial vaginitis in American women between the ages of 14 and 49 years was approximately 29% from 2001 to 2004 [4]. BV can also lead to complications such as a heightened susceptibility to sexually transmitted diseases, encompassing HIV, pelvic inflammatory disease, preterm birth, and postoperative infections [5–12]. Moreover, after existing treatments, BV still has a high recurrence rate [13, 14].

Globally, pollutants such as heavy metals exist in air, water, food, and soil, which explains their widespread exposure to humans [15]. It has been shown in the literature that heavy metal exposure causes great harm to human immunity, endocrine function, microbial communities, and other functions [16–22]. Empirical evidence suggests that exposure to heavy metals, specifically cadmium (Cd) and lead (Pb), may have detrimental effects on immune function, potentially leading to an increased susceptibility to infections [23–26]. A cross-sectional analysis conducted among non-smokers demonstrated that aside from causing harm to humans, heavy metals are similar to antibiotics and can also lead to an imbalance in the microbiota, which can negatively impact our bodies [27]. Evidence suggests that heavy metals like Pb and Cd can disrupt the equilibrium of human reproductive hormones, which will contribute to the reduction of fertility or increase the risk of endocrine-secreting cancers or several side effects [28–30].

Currently, it has been determined that BV is a condition where dysbiosis is present in the typical bacterial population found in the vagina [31]. The abnormal operation of vaginal mucosal immune [32] and hormonal milieu [33] in women take part in the vaginal microbiota dysbiosis that participates in the occurrence of BV. Serum exposure to heavy metals can indirectly affect the immune and endocrine function of the human body by affecting the microbiome [34, 35] or directly affecting the immune and endocrine function of women, thereby increasing the risk of BV in theory [36, 37]. Nonetheless, more empirical research is needed to examine the direct correlation between serum heavy metal exposure and BV. Consequently, the primary intention of this paper is to explore the correlation of serum heavy metal exposure with the danger of BV by utilizing data from the National Health and Nutrition Examination Survey (NHANES) from 2001 to 2004 among U.S. females. In the meantime, to reduce bias and make the results more scientifically robust, the potential confounders based on their associations with the outcomes of interest or a change in effect estimate of more than 10% were selected in the analysis. Covariates were included as potential confounders in the final analysis models if they changed the estimates of heavy metals on BV by more than 10% or were significantly associated with BV [38, 39].

## 2. Methods

### 2.1 Ethics approval and consent to participate

The authors bear responsibility for all facets of the work, making sure that any doubts regarding the precision or comprehensiveness of any research element are properly examined and answered. The Institutional Research Ethics Review Board of the National Center for Health Statistics of the Centers for Disease Control and Prevention authorized each protocol for the

NHANES. The study has been approved by the Ethics Review Board of the NCHS. Written informed permission was acquired by each subject. The website https://www.cdc.gov/nchs/nhanes/about_nhanes.htm has all further information.

## 2.2 Data sources

The source of our data was selected from the NHANES database (https://www.cdc.gov/nchs/nhanes/about_nhanes.htm), a study that focused on the health topics of the general populations of all ages in the United States. The NHANES is an investigation programme designed to assess the health and nutritional status of adults and children in the United States of America. The survey analyses a nationally representative sample of approximately 5,000 individuals on an annual basis. The sample for the survey is selected in such a way as to be representative of the population of the United States, encompassing all age groups (https://www.cdc.gov/nchs/nhanes/about_nhanes.htm). The study was approved by the NCHS Ethics Review Board.

The flowchart outlining the exclusion criteria is shown in Fig 1. The data of this study were obtained from two cycles of NHANES, 2001–2002 and 2003–2004. Our research examined a total of 21,161 individuals who had the potential to be suitable participants and had completed both the interview and the Mobile Examination Centers (MECs) examination. Furthermore, we excluded individuals with incomplete information regarding BV (n = 18,355) and serum heavy metal levels (n = 99). Following the exclusion of populations lacking crucial covariate data which included age, body mass index (BMI), Poverty Income Ratio (PIR), HDL cholesterol (High-Density Lipoprotein), and so on, a total of 2,493 18 years to 49 years female participants were ultimately enrolled in our research. The population analyzed corresponds to 92% of the eligible population. These data can be accessed for free on the Internet through the NHANES data files.

## 2.3 Diagnosis of BV

It was easy for us to find the measurement procedures and diagnosis of BV in the NHANES documentation [40–42]. The MECs of NHANES collected samples of vaginal swabs from adult female participants aged 18–49. After being obtained, these swabs were placed onto pH paper, ensuring that any collected samples were effectively preserved. Once the samples were secured onto the pH paper, they were attached to slides using delicate yet precise techniques. Subsequently, these slides were subjected to Gram staining. Bacterial vaginitis was defined strictly by determining the Nugent score, which was most often used in assessing the situation of Gram-stained vaginal secretions. Nugent's approach investigated the decrease in lactobacilli and gram-positive rods, which are considered part of the normal flora, as well as the decline in gram-variable morphotypes and the presence of gram-negative bacteria associated with BV [43]. The scores assigned as 0–6 were regarded as within the normal range, whereas a score of 7–10 indicated a positive outcome that suffered from BV.

## 2.4 Measurements of serum heavy metals

According to the NHANES website, we collected serum heavy metals data, including total mercury, cadmium, and lead. The website comprehensively explains the laboratory methods, elaborating on the specific techniques and procedures used for conducting experiments. Serum total mercury, lead, and cadmium were measured by the Centers for Disease Control and Prevention National Center for Environmental Health. The quantification measure of Pd and Cd were detected with a PerkinElmer SIMAA 6000 simultaneous multi-element atomic absorption spectrometer with Zeeman background correction and Hg was determined by flow injection cold vapor atomic absorption analysis with online microwave digestion in 2001–

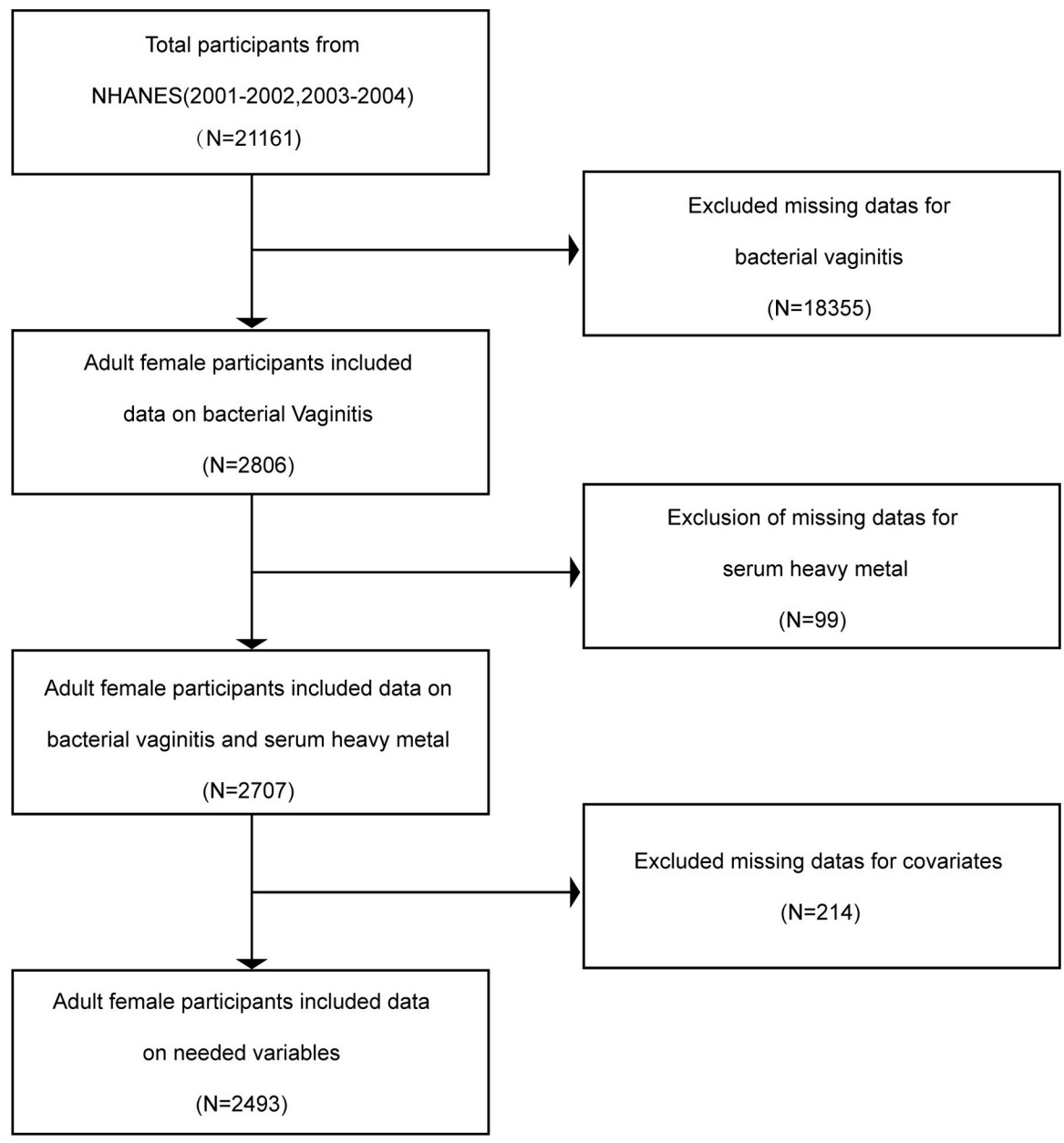

**Fig 1. Flow chart of participants' recruitment.**

2002. Inductively coupled plasma–mass spectrometry (ICP–MS) was utilized to ascertain the concentrations of Hg, Pb, and Cd in the entire blood in 2003–2004. Results below the lower limit of detection are substituted with a value equal to the limit of detection divided by the square root of two.

## 2.5 Covariates

To reduce the bias of the outcome and the occurrence of false-positive and ensure the authenticity and reliability of the findings in the study [39, 44, 45], we selected the confounders based on their associations with the outcomes of interest or a change in effect estimate of more than

10%. Covariates were included as potential confounders in the final models if they changed the estimates of heavy metals on BV by more than 10% or were significantly associated with BV [38, 39]. Considering the two criteria, participants' age, BMI, PIR, HDL cholesterol [46], total cholesterol [47, 48], uric acid [49], total calcium [48], race, education, marital status [50, 51], total number of people in the household, vigorous activity over the past 30 days and moderate exercise over the past 30 days [52] were selected as our study's covariates. In order to determine an individual's BMI, it is imperative to divide their weight, which is quantified in kilograms, by the square of their height, measured in meters. The study classified participants into three racial groups: non-Hispanic Black, non-Hispanic White, and individuals of other ethnic. The marital status of the respondents was categorized as married, divorced, widowed, separated, never married, or living with a partner. Education level was categorized into three groups: individuals with less than high school, high school diploma, and education beyond high school. Results for HDL cholesterol, total cholesterol, uric acid, and complete calcium were extracted from the NHANES laboratory database.

## 2.6 Statistical analysis

EmpowerStats software (https://www.empowerstats.net/cn/) and the statistical package R (www.r-project.org) were utilized to establish all analyses. With the aim of ascertaining the statistical significance of the investigation, a threshold of $p < 0.05$ was considered. Descriptive analysis was conducted on continuous variables through mean, standard deviation, and categorical variables through weighted percentages (%). The relationship between clinical outcome and exposure was reflected by the odds ratio (OR) value [53]. In order to obtain a more precise assessment of the outcome, a 95% confidence interval was computed.

The correlation between serum heavy metals and BV was examined using a multivariable logistic regression analysis. We constructed three models, and the details are as follows. The unadjusted model did not incorporate any modifications to the variables. Demographic variables were considered to adjust in model I, which included age, race, BMI, and education level. Model II underwent a comprehensive adjustment process, encompassing all covariates that were included. Meanwhile, tertiles of serum heavy metals levels were established in accordance with the distribution of the investigated population.

The potential nonlinearity was resolved by employing three smooth curve fittings. Aiming to alleviate potential bias in the study, stratified analyses were generated with the use of covariates. Stratified analyses were carried out to account for the confounding factors of age, race, education, marital status [54], BMI [55], HDL -cholesterol, total calcium, and uric acid. In accordance with the WHO classification of obesity status, the study population was divided into 6 groups on the basis of BMI [56, 57]: <18.5(underweight), 18.5–24.9(normal), 25.0–29.9 (overweight), ≥30.0(obesity). Additionally, the research categorized marital status as married, widowed and divorced, separated, never married, and living with partner. Sensitivity analysis was performed after the missing data of covariates were populated by multiple interpolations when the missing data of heavy metals and BV were eliminated.

## 3. Result

### 3.1 Baseline characteristics of the research participants

According to BV, Table 1 lists the descriptive features of the participants. Participants with higher BMI, serum lead, serum cadmium, total cholesterol, uric acid, and total calcium levels, lower PIR and HDL-cholesterol levels, belonging to non-Hispanic black and other racial backgrounds, having received education below a high school level, lacking vigorous and moderate physical activity in the past 30 days, and being unmarried, were found to have a higher

**Table 1. Baseline characteristics of the study population by BV status.**

| | | Bacterial vaginosis (BV) | | |
|---|---|---|---|---|
| | Overall (N = 2493) | Negative (Nugent-BV≤6) (N = 1691) | Positive (Nugent-BV≥7) (N = 802) | P-value |
| Age(year), Mean ± SD | 31.29 ± 9.72 | 31.36± 9.57 | 31.13 ± 10.04 | 0.581 |
| BMI (kg/m^2)[a], Mean ± SD | 28.13 ± 7.25 | 27.61 ± 6.98 | 29.22 ± 7.70 | <0.001 |
| Lead(ug/dl), Mead ± SD | 1.24 ± 1.19 | 1.16 ± 1.07 | 1.42 ± 1.41 | <0.001 |
| Total mercury (ug/l), Mean ± SD | 1.31 ± 1.78 | 1.31 ± 1.78 | 1.26 ± 1.70 | 0.366 |
| Cadmium (ug/l), Mean ± SD | 0.50 ± 0.54 | 0.46 ± 0.50 | 0.60 ± 0.61 | <0.001 |
| PIR, Mean ± SD | 2.38 ± 1.65 | 2.54 ± 1.68 | 2.06 ± 1.55 | <0.001 |
| HDL-cholesterol (mg/dl), Mean ± SD | 57.04±15.70 | 57.72 ± 15.56 | 55.61 ± 15.89 | 0.002 |
| Total cholesterol (mg/dl), Mean ± SD | 192.59±42.08 | 193.80 ± 41.55 | 190.02 ± 43.08 | 0.036 |
| Uric acid (mg/dl), Mean ± SD | 4.34 ± 1.04 | 4.26 ± 1.00 | 4.49 ± 1.09 | <0.001 |
| Total calcium (mg/dl), Mean ± SD | 9.36 ± 0.37 | 9.35 ± 0.37 | 9.39 ± 0.38 | 0.010 |
| Race, N (%) | | | | <0.001 |
| Non-Hispanic White | 1115 (44.73%) | 872 (51.57%) | 243 (30.30%) | |
| Non-Hispanic Black | 576 (23.11%) | 280 (16.56%) | 296 (36.91%) | |
| Other | 802 (32.17%) | 539 (31.88%) | 263 (32.79%) | |
| Education, N (%) | | | | <0.001 |
| Less Than High School | 656 (26.31%) | 391 (23.12%) | 265 (33.04%) | |
| High School Diploma (including GED) | 596 (23.91%) | 389 (23.00%) | 207 (25.81%) | |
| More Than High School | 1241 (49.78%) | 911 (53.87%) | 330 (41.15%) | |
| Marital status, N (%) | | | | <0.001 |
| Married | 1174 (47.09%) | 887 (52.45%) | 287 (35.79%) | |
| Widowed | 25 (1.00%) | 16 (0.95%) | 9 (1.12%) | |
| Divorced | 166 (6.66%) | 97 (5.74%) | 69 (8.60%) | |
| Separated | 87 (3.49%) | 45 (2.66%) | 42 (5.24%) | |
| Never married | 813 (32.61%) | 502 (29.69%) | 311 (38.78%) | |
| Living with partner | 228 (9.15%) | 144 (8.52%) | 84 (10.47%) | |
| Total number of people in the household, N (%) | | | | 0.312 |
| 1 | 142 (5.70%) | 100 (5.91%) | 42 (5.24%) | |
| 2 | 497(19.94%) | 345 (20.40%) | 152 (18.95%) | |
| 3 | 533(21.38%) | 375 (22.18%) | 158 (19.70%) | |
| 4 | 548(21.98%) | 371 (21.94%) | 177 (22.07%) | |
| 5 | 418(16.77%) | 273 (16.14%) | 145 (18.08%) | |
| 6 | 147 (5.90%) | 90 (5.32%) | 57 (7.11%) | |
| 7 or more people in the Household | 208 (8.34%) | 137 (8.10%) | 71 (8.85%) | |
| Vigorous activity over the past 30 days, N (%) | | | | <0.001 |
| Yes | 881(35.340%) | 646 (38.20%) | 235 (29.30%) | |
| No | 1612(64.66%) | 1045 (61.80%) | 567 (70.70%) | |
| Moderate activity over the past 30 days, N (%) | | | | 0.010 |
| Yes | 1380(55.36%) | 966 (57.13%) | 414 (51.62%) | |
| No | 1113(44.65%) | 725 (42.87%) | 388 (48.38%) | |

BMI was calculated as the body weight in kilograms divided by the square of the height in meters.

P value: if it is a continuous variable, it is obtained by the Kruskal Wallis rank sum test. If the theoretical number of counting variables is less than 10, it is received by the Fisher exact probability test.

% for categorical variables: P value was calculated by chi-square test.

Abbreviations: BMI, Body Mass Index; PIR, Poverty Income Ratio; HDL, High-Density Lipoprotein; GED, General educational development.; OR odds ratio; CI, confidence interval.

likelihood of testing positive for BV compared to those who tested negative. Significant differences were not discovered in age, serum total mercury levels, and household size (p >0.05).

## 3.2 Correlation between serum lead, total mercury, cadmium, and BV

The association between the collective and the three distinct categories of serum lead, total mercury, and cadmium concentrations and BV is presented in Table 2. There was a notable positive association between the overall concentration of lead in the serum and the likelihood of developing BV, resulting in an 11% increase in the danger of BV [OR = 1.11 (1.02,1.19), p = 0.012]. The group with the highest lead levels in the serum exhibited the highest incidence of developing BV in comparison to the remaining two lower groups, exhibiting a statistically significant 35% rise in the possibility of disease development, as opposed to the group with low levels [OR = 1.35 (1.06,1.72), p = 0.016]. In all three models, a positive correlation between serum lead levels from 1.3 to 23.9 µg/dl and the occurrence of BV was significant and stable. The correlation between serum lead levels and the occurrence of BV was intuitively described using smoothing curves, as depicted in Fig 2 (p = 0.020).

The study found that there was a passive association between serum total mercury levels ranging from 0.5 to 1.1 µg/l and a 2% decrease in the risk of BV, and the levels from 1.2 to 27.4 µg/l were negatively correlated with a 4% decrease in the risk of BV in model II. Nevertheless, the research findings did not demonstrate statistical significance(p>0.05). There was no statistically significant association found between overall serum total mercury levels and the risk of BV (p>0.05).

**Table 2. Multivariable-adjusted generalized linear regression models for association among heavy metal exposure and bacterial vaginosis.**

|  | Unadjusted Model | Model I | Model II |
|---|---|---|---|
| Outcome | OR (95%Cl) P value | OR (95%Cl) P value | OR (95%Cl) P value |
| Serum lead (ug/dl) |  |  |  |
| Overall | 1.20 (1.11, 1.30) <0.001 | 1.14 (1.05, 1.23) 0.001 | 1.11 (1.02, 1.19) 0.012 |
| Q1 (0.2 to 0.7) | 1.0 [Reference] | 1.0 [Reference] | 1.0 [Reference] |
| Q2 (0.8 to 1.2) | 1.24 (1.01, 1.54) 0.045 | 1.17 (0.94, 1.47) 0.166 | 1.10 (0.88, 1.39) 0.402 |
| Q3 (1.3 to 23.9) | 1.72 (1.39, 2.11) <0.001 | 1.48 (1.17, 1.87) 0.001 | 1.35 (1.06, 1.72) 0.016 |
| P for trend | <0.001 | 0.001 | 0.015 |
| Serum total mercury (ug/l) |  |  |  |
| Overall | 0.98 (0.93, 1.03) 0.367 | 0.99 (0.94, 1.04) 0.593 | 1.00 (0.94, 1.05) 0.889 |
| Q1 (0.07 to 0.4) | 1.0 [Reference] | 1.0 [Reference] | 1.0 [Reference] |
| Q2 (0.5 to 1.1) | 1.11 (0.90, 1.37) 0.334 | 1.01 (0.81, 1.26) 0.952 | 1.02 (0.82, 1.28) 0.847 |
| Q3 (1.2 to 27.4) | 1.00 (0.81, 1.25) 0.969 | 0.92 (0.73, 1.17) 0.486 | 0.96 (0.75, 1.23) 0.763 |
| P for trend | 0.949 | 0.460 | 0.741 |
| Serum cadmium (ug/l) |  |  |  |
| Overall | 1.55 (1.33, 1.81) <0.001 | 1.63 (1.38, 1.92) <0.001 | 1.48 (1.25, 1.76) <0.001 |
| Q1 (<0.2) | 1.0 [Reference] | 1.0 [Reference] | 1.0 [Reference] |
| Q2 (0.2 to 0.4) | 0.99 (0.74, 1.34) 0.962 | 0.94 (0.69, 1.28) 0.686 | 0.98 (0.71, 1.35) 0.912 |
| Q3 (0.5 to 7.4) | 1.54 (1.13, 2.10) 0.007 | 1.49 (1.07, 2.08) 0.017 | 1.41 (1.01, 1.98) 0.047 |
| P for trend | <0.001 | <0.001 | 0.002 |

Unadjusted model: no covariates were adjusted.

Model I: Part of covariates, age, race, education, and BMI were adjusted.

Model II: All covariates, age, race, education, BMI, moderate activity over the past 30 days, PIR, marital status, HDL-cholesterol, total cholesterol, total number of people in the household, total calcium, uric acid, and vigorous activity over the past 30 days were adjusted.

Abbreviations: BMI, Body Mass Index; PIR, Poverty Income Ratio; HDL, High-Density Lipoprotein; CI, confidence interval; OR, odds ratio.

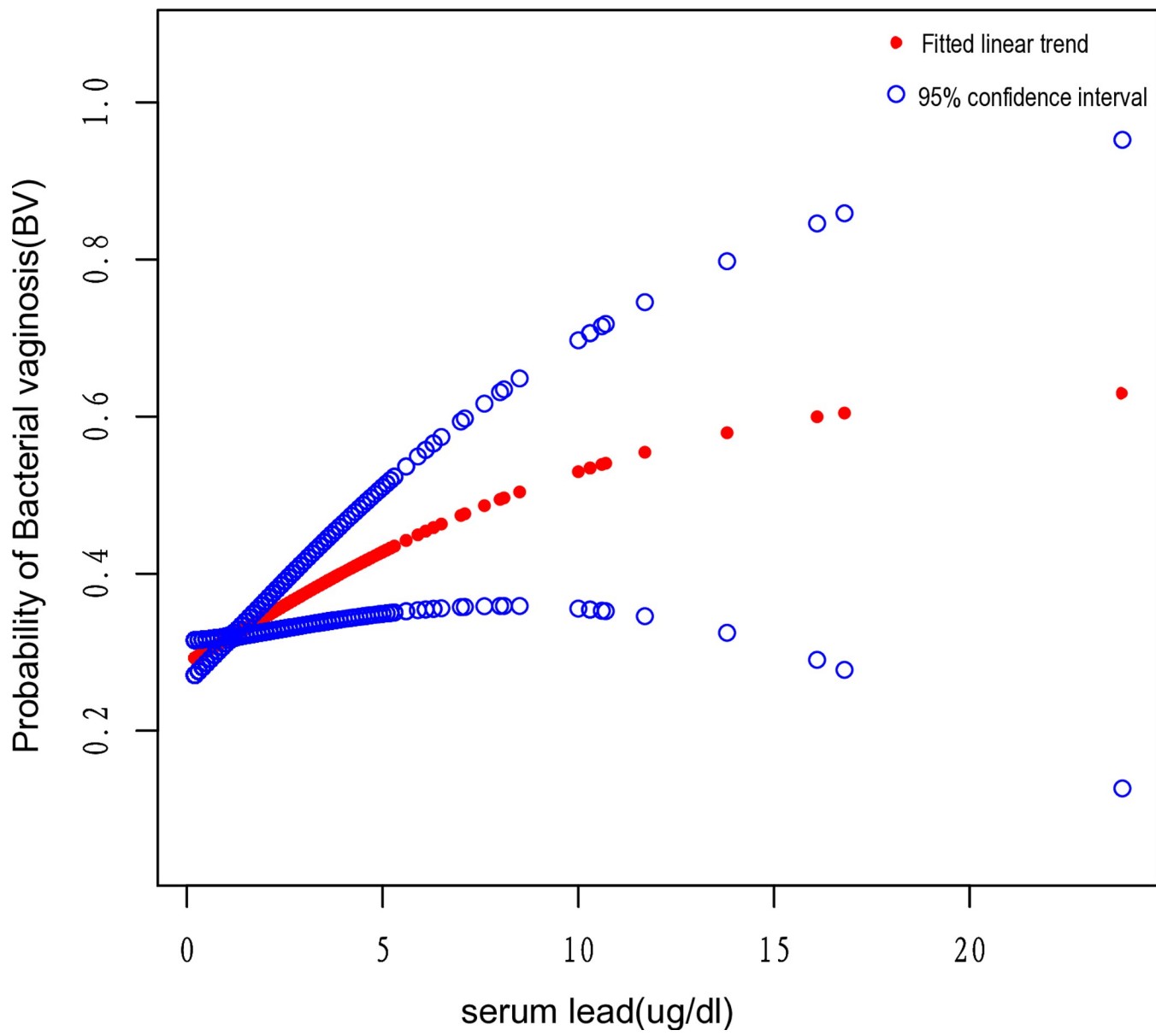

**Fig 2. Correlation between serum lead and risk of BV.** The association between serum lead and the risk of BV was exhibited by a smooth curve fitting model. The central red dots represent serum lead concentrations, with each point contributing to a continuous fitted curve. The region between the two blue dashed lines corresponds to the 95% confidence interval. The X-axis is serum lead levels (continuous variable), and the Y-axis is odds ratios (ORs). ORs were computed from the Model II in a multivariate logistic regression analysis.

There existed a clear correlation between the collective concentration of cadmium in the serum and the likelihood of developing BV, as indicated by a 48% rise in the risk of BV when adjusted for all covariates in the model [OR = 1.48 (1.25,1.76), p<0.001]. The serum cadmium levels ranging from 0.5 to 7.4 µg/l exhibited a 41% rise in the susceptibility to BV in model II, indicating a direct correlation with the risk of BV and statistical significance [OR = 1.41 (1.01,1.98), p = 0.047]. The positive relationship between serum cadmium and the risk of BV was steady in all three models. Fig 3 demonstrates the correlation between serum cadmium and the risk of BV intuitively through smoothing curves(p<0.001).

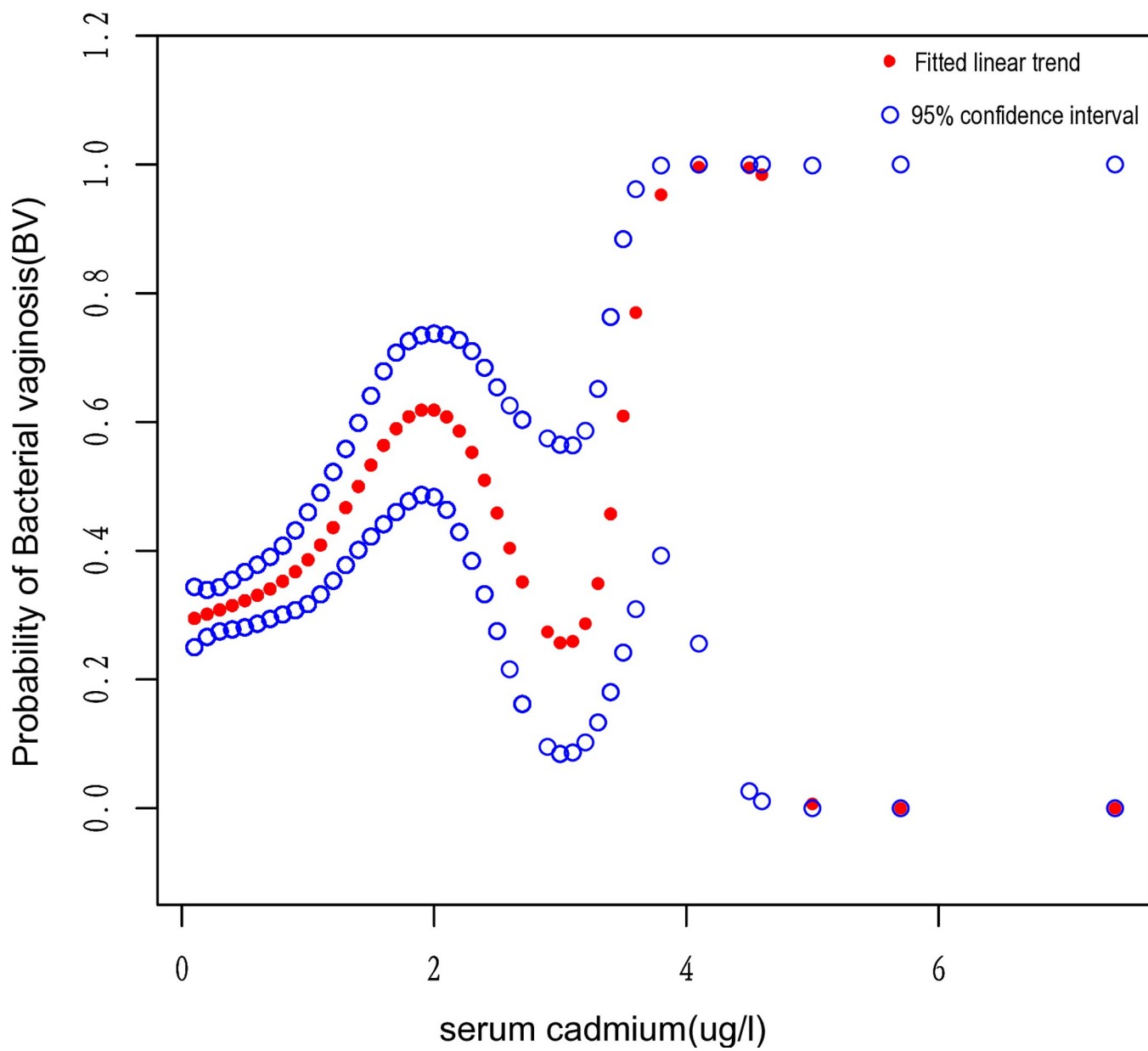

**Fig 3. Correlation between serum cadmium and risk of BV.** The association between serum cadmium and the risk of BV was exhibited by a smooth curve-fitting model. The central red dots represent serum cadmium concentrations. The region between the two blue dashed lines corresponds to the 95% confidence interval. The X-axis is serum cadmium levels (continuous variable), and the Y-axis is odds ratios (ORs). ORs were computed from the Model II in a multivariate logistic regression analysis.

### 3.3 Stratification analysis between serum lead, total mercury, cadmium, and BV

Table 3 describes the stratification analysis for age, race, education, marital status, BMI, HDL-cholesterol, total calcium, and uric acid between serum lead, total mercury, and cadmium and the risk of BV. In the case of serum lead and cadmium, the results of our stratified analysis reflected that the direction of the result was mostly consistent (OR>1). For Pb, the risk further increases in females who were 37 to 49 years old, accepted education less than high school,

**Table 3. Association between serum heavy metal exposure and Bacterial vaginosis in different subgroups.**

| Subgroup | Serum lead(ug/dl) OR (95%Cl) P value | Serum total mercury (ug/l) OR (95%Cl) P value | Serum cadmium(ug/l) OR (95%Cl) P value |
|---|---|---|---|
| **Age(years)** | | | |
| **18 to 24 (n = 804)** | 1.02 (0.91, 1.15) 0.738 | 1.05 (0.95, 1.15) 0.326 | 2.03 (1.38, 2.99) <0.001 |
| **25 to 36 (n = 857)** | 1.13 (0.95, 1.35) 0.171 | 0.86 (0.76, 0.98) 0.022 | 1.32 (0.93, 1.86) 0.119 |
| **37 to 49 (n = 832)** | 1.20 (1.05, 1.37) 0.008 | 1.01 (0.93, 1.10) 0.770 | 1.40 (1.10, 1.78) 0.006 |
| **Race** | | | |
| **Non-Hispanic White (n = 1115)** | 1.15 (0.96, 1.37) 0.138 | 0.96 (0.88, 1.06) 0.419 | 1.34 (1.07, 1.67) 0.011 |
| **Non-Hispanic Black (n = 576)** | 1.11 (0.93, 1.32) 0.258 | 1.00 (0.89, 1.12) 0.974 | 1.63 (1.07, 2.47) 0.022 |
| **Other (n = 802)** | 1.10 (0.99, 1.21) 0.073 | 1.05 (0.96, 1.16) 0.268 | 1.42 (0.93, 2.17) 0.102 |
| **Education** | | | |
| **Less Than High School (n = 656)** | 1.14 (1.02, 1.28) 0.020 | 1.04 (0.94, 1.15) 0.475 | 1.87 (1.34, 2.59) <0.001 |
| **High School Diploma (including GED) (n = 596)** | 1.24 (0.97, 1.60) 0.088 | 1.04 (0.93, 1.16) 0.483 | 2.02 (1.44, 2.85) <0.001 |
| **More Than High School (n = 1241)** | 1.01 (0.87, 1.17) 0.907 | 0.95 (0.88, 1.03) 0.243 | 1.07 (0.81, 1.41) 0.624 |
| **Marital status** | | | |
| **Married (n = 1174)** | 1.20 (1.05, 1.38) 0.008 | 0.96 (0.88, 1.06) 0.408 | 1.13 (0.84, 1.52) 0.426 |
| **Widowed and Divorced (n = 191)** | 1.03 (0.80, 1.31) 0.840 | 1.03 (0.84, 1.25) 0.801 | 2.40 (1.50, 3.83) <0.001 |
| **Separated (n = 87)** | 2.47 (1.03, 5.91) 0.043 | 1.04 (0.56, 1.92) 0.910 | 1.21 (0.66, 2.22) 0.534 |
| **Never married (n = 813)** | 1.07 (0.89, 1.28) 0.462 | 1.06 (0.97, 1.15) 0.174 | 2.03 (1.36, 3.02) <0.001 |
| **Living with partner (n = 228)** | 1.06 (0.92, 1.22) 0.422 | 0.77 (0.57, 1.04) 0.091 | 1.23 (0.72, 2.09) 0.450 |
| **BMI (kg/m^2)** | | | |
| **15.09 to 18.5 (n = 79)** | 1.12 (0.65, 1.93) 0.676 | 1.14 (0.40, 3.20) 0.877 | 2.69 (0.46, 15.62) 0.271 |
| **18.51 to 24.90 (n = 900)** | 1.00 (0.89, 1.13) 0.943 | 1.05 (0.96, 1.14) 0.268 | 1.58 (1.15, 2.17) 0.005 |
| **24.91 to 29.90 (n = 686)** | 1.14 (0.98, 1.33) 0.092 | 0.93 (0.82, 1.04) 0.210 | 1.60 (1.11, 2.29) 0.011 |
| **29.91 to 65.41 (n = 828)** | 1.27 (1.06, 1.53) 0.010 | 0.98 (0.88, 1.10) 0.746 | 1.32 (1.01, 1.73) 0.042 |
| **HDL-cholesterol (mg/dl)** | | | |
| **23 to 47 (n = 761)** | 1.13 (0.98, 1.30) 0.095 | 1.01 (0.92, 1.11) 0.821 | 1.20 (0.91, 1.56) 0.193 |
| **48 to 61 (n = 888)** | 1.02 (0.91, 1.15) 0.719 | 1.01 (0.92, 1.12) 0.835 | 1.79 (1.33, 2.41) <0.001 |
| **62 to 160 (n = 844)** | 1.19 (1.01, 1.41) 0.042 | 0.96 (0.87, 1.06) 0.428 | 1.50 (1.04, 2.16) 0.031 |
| **Total calcium (mg/dl)** | | | |

*(Continued)*

**Table 3.** (Continued)

| Subgroup | Serum lead(ug/dl) OR (95%Cl) P value | Serum total mercury (ug/l) OR (95%Cl) P value | Serum cadmium(ug/l) OR (95%Cl) P value |
|---|---|---|---|
| 8.1 to 9.1 (n = 691) | 1.12 (0.98, 1.28) 0.097 | 0.93 (0.81, 1.07) 0.308 | 1.85 (1.26, 2.72) 0.002 |
| 9.2 to 9.4 (n = 794) | 1.24 (1.05, 1.46) 0.014 | 1.13 (1.02, 1.25) 0.024 | 1.13 (0.85, 1.52) 0.398 |
| 9.5 to 11.3 (n = 1008) | 1.05 (0.92, 1.20) 0.449 | 0.92 (0.84, 1.02) 0.107 | 1.70 (1.29, 2.22) <0.001 |
| Uric acid (mg/dl) | | | |
| 1.7 to 3.7 (n = 768) | 1.00 (0.88, 1.14) 0.977 | 1.05 (0.96, 1.15) 0.298 | 1.34 (0.93, 1.92) 0.114 |
| 3.8 to 4.6 (n = 853) | 1.19 (1.04, 1.37) 0.011 | 0.93 (0.84, 1.04) 0.192 | 1.89 (1.39, 2.57) <0.001 |
| 4.7 to 8.1 (n = 872) | 1.19 (1.02, 1.38) 0.024 | 1.00(0.91, 1.10) 0.979 | 1.30 (1.00, 1.70) 0.051 |

The logistic regression models constructed for subgroup analyses were adjusted for all covariates: age(excluded in analysis stratified by age), race(excluded in analysis stratified by race), education(excluded in analysis stratified by education status), BMI(excluded in analysis stratified by BMI), moderate activity over the past 30 days, PIR, marital status(excluded in analysis stratified by marital status), HDL-cholesterol (excluded in analysis stratified by HDL-cholesterol), total cholesterol, total number of people in the household, total calcium(excluded in analysis stratified by total calcium), uric acid(excluded in analysis stratified by uric acid) and vigorous activity over the past 30 days. Abbreviations: BMI, Body Mass Index; PIR, Poverty Income Ratio; HDL, High-Density Lipoprotein; GED, General educational development CI, confidence interval; OR, odds ratio.

married and separated, BMI from 29.91 to 65.41 kg/m^2, HDL-cholesterol from 62 to 160 mg/dl, total calcium 9.2 to 9.4 mg/dl, and uric acid from 3.8 to 8.1 mg/dl. In addition, for Cd, the risk further increased in females who were 18 to 24 years old and 37 to 49 years old, non-Hispanic white and non-Hispanic black, accepted education less than high school, and high school diploma (including GED), widowed and divorced, never married, BMI from 18.51 to 65.41 kg/m^2, HDL-cholesterol from 48 to 160 mg/dl, total calcium from 8.1 to 9.1 mg/dl and 9.5 to 11.3 mg/ml, and Uric acid from 3.8 to 4.6 mg/dl.

Females aged 25 to 36 exhibited statistically significant findings regarding the correlation between serum total mercury and the risk of BV (OR: 0.86, 95% CI: 0.76–0.98). However, the other result of the study was unstable and insignificant, and stratified analysis showed that OR fluctuations were more pronounced in different subgroups. The correlation between serum total mercury and the risk of BV among females aged between 25 and 36 is shown in Fig 4 (p = 0.104).

### 3.4 Sensitivity analysis

Sensitivity analysis was conducted when the missing covariates data had been filled in the blank by multiple interpolations. The orientation of the sensitivity analysis results (S1 Table) is also generally coherent with the former results.

## 4. Discussion

As far as we are aware, the cross-sectional analysis marked the inaugural investigation into the association between serum heavy metals and BV, utilizing comprehensive data from NHANES

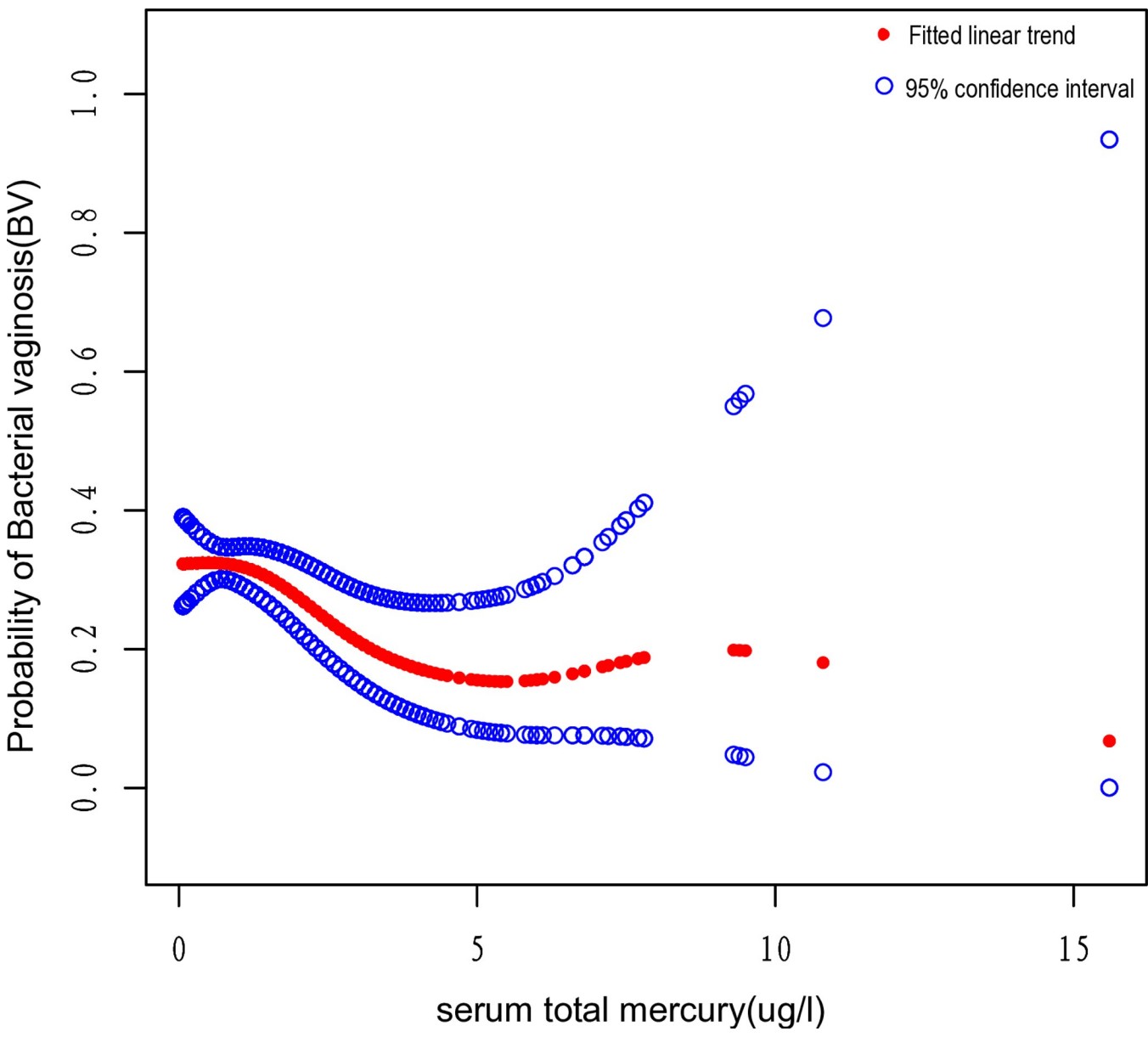

**Fig 4. Correlation between serum total mercury and risk of BV among females aged between 25 and 36.** The association between serum total mercury and the risk of BV among females aged between 25 and 36 was exhibited by a smooth curve fitting model. The central red dots represent serum total mercury concentrations, with each point contributing to a continuous fitted curve. The region between the two blue dashed lines corresponds to the 95% confidence interval. The X-axis is serum total mercury levels (continuous variable), and the Y-axis is odds ratios (ORs). ORs were computed from the Model II in a multivariate logistic regression analysis.

spanning the years 2001 to 2004. The consequence of multivariable-adjusted generalized logistic regression models demonstrated significant positive correlations between serum Pb and Cd, and the risk of BV. The stratification analysis and sensitivity analysis confirmed that the results were reliable and steady. In contrast, expect a conspicuous negative correlation between serum Hg and BV among females aged 25 to 36 in stratification analysis, the research did not reveal a significant correlation between the serum Hg and the occurrence of BV. Consequently, ameliorating serum heavy metal status in females may constitute a biological substrate for the clinical reduction of BV infection and the prevention of recurrence.

Among adults, occupational exposure is the most frequent cause of high heavy metal levels. Additionally, being in touch with cigarette smoking, food, dental amalgam, and air are the primary sources of multiple toxic metals [58], which are omnipresent environmental contaminants and could accumulate in humans and other organisms with long half-lives [59]. Recently, research has demonstrated that exposure to heavy metals like Pb, Cd, and Hg which are especially considered endocrine-disrupting compounds (EDCs) have been indicated to be positively associated with the risk of diseases of the reproductive system. A case-control study has suggested that exposure to multiple toxic metals involving Pb, Hg, As, Ba, and Cd was in connection with the increased occurrence of polycystic ovarian syndrome [60]. Moreover, Pb and Cd were denoted as metalloestrogens were participated in the pathogenesis of hormone-related cancers like ovarian, endometrial, and cervical cancer through the generation of oxidative stress and DNA damage [61]. Alternatively, chronic exposure to heavy metals has been linked to an increased risk of developing breast cancer, endometriosis, endometrial cancer, menstrual disorders, and spontaneous abortions, as well as preterm deliveries and stillbirths [62]. In summary, environmental exposure to heavy metals has a detrimental impact on female reproductive health.

Various putative and potential biological mechanisms may account for the important role of heavy metals in the risk of BV. BV, as a prevalent inflammatory disease occurring in the female reproductive system, is distinguished by an imbalance in the vaginal microbiota [63]. It is becoming increasingly clear that BV is caused by anaerobes etc. at least 35 bacterial phylotypes overpowering the Lactobacillus species that are the dominant species in healthy vaginal flora [64]. Previous studies have shown that hormonal level plays a significant role in maintaining gynecological health and hormonal level perturbations can result in imbalances in vaginal microbiota [65]. Heavy metals have been implicated as EDCs by interfering with hormone receptors [66] and disrupting the circadian pattern of hypothalamic-pituitary-gonadal (HHG) axis activity [67, 68] to contribute to disruption of the endocrine system and hormonal level perturbations. In addition, previous research has demonstrated that female mucosal immunity, involving vaginal epithelial cells and local lymphoid tissue, plays an important role in the prevention of BV [69]. Barrier defense, involving epithelial and endothelial cells, is an important component of innate immunity. In parallel, it is of the utmost importance to protect the mucosal surfaces of the female vaginal tract through innate and adaptive immunological mechanisms [70]. It is commonly acknowledged that heavy metals are immunotoxic inhibitors that damage the immune system to weaken the organism's resistance and turn the symptoms on, including innate and adaptive immunity, which occurs in a time- and dose-dependent manner [36, 71, 72].

Ultimately, no direct evidence suggests a connection between heavy metals and the likelihood of BV. In our study, to provide a more comprehensive and intuitive demonstration of the study results, we employed various methodological adjustments such as hierarchical analysis, confounding, multiple regression analysis, and curve fitting to achieve this. These adjustments allowed us to delve deeper into the research and further explore the relationship. The findings that heavy metals are positively associated with BV could provide regulatory agencies with the ability to revise guidelines for heavy metals, which can be used to guide clinically effective prevention strategies and initiate targeted treatment options.

In contrast, a few unavoidable restrictions to the present study are worth noting. First, the cross-sectional observational study cannot establish a causal relationship between exposure to heavy metals and the likelihood of developing BV and reveal the pathways between heavy and BV. Second, as the duration of decay for each metal present in the human body exhibits variability, the existing method could not accurately gauge the concentration of metals in the individual's body. Third, we consulted multiple papers and included as many recognized

confounders as possible in the covariates. However, limited to the restricted type of data in the NHANES database, we were unable to exclude all confounders. Finally, due to the limitation of the collection of samples of BV in the NHANES, data were collected from 2001 to 2004, which represented the health condition at that time and it is essential to note that this specific duration might introduce a certain degree of bias when trying to extrapolate the findings to the current environment and other races. The utilization of historical data may lead to a distortion of trends, as the data set is limited in temporal scope and may fail to capture significant trends and changes that occur over extended periods. In summary, further investigation and clarification of the precise relationship between heavy metals and BV may require prospective cohort studies involving large sample sizes and the implementation of mechanistic studies.

## 5. Conclusion

The cross-sectional study unveiled a substantial correlation between serum Pb and Cd with elevated susceptibility to BV. At the same time, Hg was identified as not significantly related to the risk of BV among females. According to the research, while there are more than a couple of heavy metals, we suggest decreasing environmental and occupational exposure to heavy metals as much as possible.

## Supporting information

**S1 Table. Sensitivity analysis of heavy metals concerning BV.**
(DOCX)

## Acknowledgments

We would appreciate the assistance of every patient and clinical researcher who gave their time and effort to our study.

## Author Contributions

**Conceptualization:** Yu-Xue Feng, Xu-Guang Guo.

**Data curation:** Yu-Xue Feng, Ming-Zhi Tan.

**Formal analysis:** Yu-Xue Feng.

**Investigation:** Yu-Xue Feng.

**Methodology:** Yu-Xue Feng.

**Supervision:** Yu-Xue Feng.

**Validation:** Yu-Xue Feng.

**Visualization:** Yu-Xue Feng.

**Writing – original draft:** Yu-Xue Feng, Hui-Han Qiu, Jie-Rong Chen, Si-Zhe Wang, Ze-Min Huang.

**Writing – review & editing:** Yu-Xue Feng, Ming-Zhi Tan.

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
