## [Decision Letter · Decision Letter 0]

9 Aug 2024

PONE-D-24-18888Association between heavy metal exposure and bacterial vaginosis: a cross-sectional studyPLOS ONE

Dear Dr. Guo, I am pleased to say that both reviewers enjoyed the manuscript very much and we are excited about the possibility of publishing your work. However, both reviewers recommended improvements (one resquest minor revisions and another major revisions) to the original manuscript. Please read carefully both reviewers’ reports addressing and answering all comments and suggestions. So, I kindly invite the authors to realize a thoughtful revision of the submitted manuscript to achieve publication endorsement by the reviewers.

Thank you and best regards,

António Machado

Reviewers' comments:

Reviewer's Responses to Questions

**Comments to the Author**

1. Is the manuscript technically sound, and do the data support the conclusions?

Reviewer #1: Partly

Reviewer #2: No

2. Has the statistical analysis been performed appropriately and rigorously? 

Reviewer #1: I Don't Know

Reviewer #2: No

3. Have the authors made all data underlying the findings in their manuscript fully available?

Reviewer #1: Yes

Reviewer #2: No

4. Is the manuscript presented in an intelligible fashion and written in standard English?

Reviewer #1: Yes

Reviewer #2: No

5. Review Comments to the Author

Reviewer #1: NHAES data were from 2001 to 2004, it appears that the study participants enrolment was more recent [ 106-107: …. were enrolled in our research]. It might lead to the supposition that data of almost two decades back on the heavy metal prevalence are being extrapolated to the current health condition. This needs clarification

If the data on BV are also obtained from the NHAES report, it may be clarified whether any earlier research on the correlation or causation of heavy metals has been done on the same dataset or not, and whether this is an entirely fresh examination. [85-88, 237- inaugural investigation]. Since the data were collected, the supposition would be that they were also examined. It may also be clarified if the earlier studies were different in their approach and results, and whether this paper does indeed present some new insights.

An inherent limitation is the pathway between the heavy metals and BV. It is not clear from the paper which of the pathways [248-251] are the most likely in this study and which need to be addressed by regulatory agencies as recommended in the paper [ 291].

Reviewer #2: The topic could be relevant in research if several points are rectified. For this reason, a thorough review of the entire study is necessary.

In the abstract the same sentences are repeated (lines 34 to 37 and 39 to 41). The abstract does not explain the importance of carrying out the study, especially of the serum heavy metal variable, nor the importance of finding the association between the two variables.

The introduction should use current literature and provide epidemiological data to support the statements made.

Reference 25 (line 75) analyzes a population of non-smokers and the results cannot be generalized to the entire population. Cite other articles or reword the sentence and change the terminology "bacterial flora" to microbiota.

Explain the meaning of "human equilibrium" (line 78).

Support lines 82 to 85 with bibliographic references.

Why were these years chosen to conduct the study and not use data from more recent years? There should be an important justification for considering the analysis of data from more than 20 years ago.

Based on what criteria are the covariates established?

Revise the total cholesterol value for the whole population (does not agree with the average obtained in the subpopulations with positive and negative BV results).

How do you justify using covariates to adjust the data? Is this data adjustment really necessary?

The figures are not clear. There are no legends explaining the meaning of the blue and red dots.

It is important to consider that the OR calculations require information on people who did and did not have exposure (heavy metals) and people who did and did not develop the event (BV). Only in this way can an association be established.

No table indicates the population with or without exposure to heavy metals and the prevalence of BV (the main association to be analyzed in this study). The purpose of analytical cross-sectional studies is to determine prevalences and none are detailed here.

The conclusion shows that causality has been found and there are results that are not even statistically significant. In addition, the possible confounders that could exist in this analysis are not analyzed.

It is necessary to determine the percentage of the population that was eligible and from which the information was obtained to analyze the data. This helps to determine that there is no over-estimation or under-estimation of the data analyzed and the results obtained. Ideally, the population analyzed should correspond to at least 80% of the eligible population.

The graphs show group data analysis, but in this study you collected information from each individual, so individual data analysis is essential since this is not an ecological study. Even the use of linear correlation analysis, which should be used only in the analysis of group data, is questionable. The use of correlation in these studies would introduce an ecological bias, because the relationship obtained at the group level may differ from what is determined at the individual level.

It is necessary to carry out an in-depth analysis of the biases that could interfere with the hypothesis established in this study. Mainly, in the confounding bias, because the existence of heavy metals in the serum could be only a confounder and this would invalidate the whole study. It is necessary to justify this section with criteria used in epidemiology.

Finally, it is necessary to carefully revise the entire text of the article, improve the wording, use current bibliography and revise the wording in English.

6. PLOS authors have the option to publish the peer review history of their article (what does this mean?). If published, this will include your full peer review and any attached files.

Reviewer #1: No

Reviewer #2: **Yes: **Sandra Pamela Cangui Panchi

---

## [Author Response · Author response to Decision Letter 0]

1 Oct 2024

Responses to reviewers' comments

Dear reviewers.

Firstly, we would like to thank you for your kind letter and the reviewers’ constructive comments concerning our article (Manuscript No.: PONE-D-24-18888). These comments are all valuable and helpful for improving our article. All the authors have seriously discussed all these comments. According to the reviewers’ comments, we have tried to modify our manuscript to meet your journal's requirements. 

In response to your suggestions, we have implemented a series of significant alterations and enhancements to the article to optimize its quality and clarity. The following is a summary of the principal modifications we have incorporated based on your feedback and implemented in the article. In addition, we have changed the author's affiliation. Point-by-point responses to the reviewers are listed below this letter.

 The specific comments from the reviewers are highlighted in red font, while my responses are in black.

To Reviewer 1

1. Data from almost two decades back on the prevalence of heavy metals may be extrapolated to the current health condition.

Thank you for your advice, data on heavy metals for this study were collected from the NHANES database from 2001 to 2004, which reflected the health condition and cannot be extrapolated to the current prevalence. We have clarified the limitation in the discussion on lines 320 to 324.

2. It may be clarified whether any earlier research on the correlation or causation of heavy metals has been done on the same dataset or not, and whether this is an entirely fresh examination. Since the data were collected, the supposition would be that they were also examined. It may also be clarified if the earlier studies differed in their approach and results, and whether this paper presents some new insights.

Thanks for the suggestion, your suggestion means a lot to us. The data on BV were obtained from the NHANES, there was no earlier research on the correlation or causation of heavy metals has been done on the same dataset and this is an entirely fresh examination. In the study, multivariable logistic regression models were utilized to assess the correlation between serum heavy metal exposure and BV, a stratified analysis was performed to further investigate the relationship among different population groups and smooth curve fittings were conducted to intuitively evaluate the correlation. The cross-sectional study unveiled a substantial correlation between serum Pb and Cd with elevated susceptibility to BV. It is the first study that demonstrated the correlation between heavy metals and the risk of BV, and this point has been emphasized in the discussion (lines 255 to 257).

3. An inherent limitation is the pathway between the heavy metals and BV. It is not clear from the paper which of the pathways are the most likely in this study and which need to be addressed by regulatory agencies as recommended in the paper.

Thank you for your valuable advice, it is a cross-sectional study that could explore the correlation between heavy metals and the risk of BV and could not reveal the pathway between the heavy metals and BV. However, women’s endocrine and immune systems may be damaged because of the presence of high concentrations of serum heavy metals and we have explained the potential pathways in the discussion. Meanwhile, we have highlighted the inherent limitation in the discussion on lines 313 to 315.

To Reviewer 2

1. The same sentences are repeated in the abstract (lines 34 to 37 and 39 to 41).

We appreciate your correlation. We are sorry for our careless mistakes. We have deleted one of the sentences to make the article more compact and improve its readability.

2. The abstract does not explain the importance of carrying out the study, especially of the serum heavy metal variable, nor the importance of finding the association between the two variables.

 Thank you for pointing out this problem in the abstract. The importance of carrying out the study, the serum heavy metal variable, and the importance of finding the association between the two variables have been explained in the introduction in detail. To improve the readability of the article, we explained the above importance briefly in the abstract (lines 43 to 45).

3. The introduction should use current literature and provide epidemiological data to support the statements made.

Thank you for your suggestions regarding our research. We have reviewed the available literature and cited relevant epidemiological studies in the introduction to support the statements made (lines 73 to 75).

4. Reference 25 (line 75) analyzes a population of non-smokers and the results cannot be generalized to the entire population. Cite other articles or reword the sentence and change the terminology "bacterial flora" to microbiota. Explain the meaning of "human equilibrium" (line 78).

Thank you so much for your careful check. We feel sorry for the improper wording. We modified the expression according to reference 26 to emphasize that the restricted population of the study was the non-smokers (lines 85 to 88). We have used “microbiota” as you suggested (lines 87). With references 27 and 28, we have also changed “human equilibrium" to “the equilibrium of human reproductive hormones” (lines 88 to 89).

5. Support lines 82 to 85 with bibliographic references.

Thank you for your review and valuable suggestions. We have support lines 82 to 85 with bibliographic references 32 to 36 (lines 93 to 96).

6. Why were these years chosen to conduct the study and not use data from more recent years? There should be an important justification for considering the analysis of data from more than 20 years ago.

Thank you for pointing out this problem in the manuscript. Due to the limitations of the NHANES database, the available data on bacterial vaginosis (BV) is only accessible for 2001-2004. Therefore, when conducting our analysis, we had to solely rely on the data collected during this specific time frame. We proceeded with the available data to perform an accurate evaluation and draw conclusions regarding BV within this limited period. This limitation has been pointed out in the limitation of the manuscript (lines 320 to 324).

7. Based on what criteria are the covariates established?

Thank you for your review. First, we reviewed the literature to include data on potential covariates. Then, according to international practice standards, we included potential covariates in the basic or full models to observe the changes in regression coefficients. Covariates were included as potential confounders in the final models if they changed the estimates of heavy metals on BV by more than 10% or were significantly associated with BV. We added the covariates in the method and listed references to support them[1, 2].

8. Revise the total cholesterol value for the whole population (does not agree with the average obtained in the subpopulations with positive and negative BV results). 

We appreciate your feedback. It is a giant mistake that affects the whole quality of our article. We feel sorry for our carelessness. We have corrected it, and we also appreciate your point. We revised the cholesterol value in the average obtained in the subpopulations with positive and negative BV results.

9. How do you justify using covariates to adjust the data? Is this data adjustment really necessary?

Thank you for your valuable feedback. We reviewed the literature to include data on potential covariates. Moreover, according to international practice standards, we included potential covariates in the basic or full models to observe the changes in regression coefficients. We selected these confounders based on their associations with the outcomes of interest or a change in effect estimate of more than 10%. Covariates were included as potential confounders in the final models if they changed the estimates of heavy metals on BV by more than 10% or were significantly associated with BV. Based on these two criteria, we screened for covariates that could affect the results to make our study as scientific as possible[1, 2].

10. The figures are not clear. There are no legends explaining the meaning of the blue and red dots. 

Thank you for your comment regarding the figure. We understand your concerns about the quality of the figures and we have re-uploaded them with greater clarity. In addition, we have added legends to the figures to explain the meaning of the blue and red dots in lines 708 to 730.

11. It is important to consider that the OR calculations require information on people who did and did not have exposure (heavy metals) and people who did and did not develop the event (BV). Only in this way can an association be established. 

Thank you for your valuable suggestion, which is indeed pertinent. In fact, our original design idea coincided with yours. However, considering that the NHANES data did not include the population with or without heavy metal exposure, there were only data measuring serum heavy metal concentrations in the population. Therefore, we only had the serum heavy metal concentration as a continuous variable to work with and designed a compartmentalized study methodology for this.

12. No table indicates the population with or without exposure to heavy metals and the prevalence of BV (the main association to be analyzed in this study). The purpose of analytical cross-sectional studies is to determine prevalences and none are detailed here. 

Thanks again for your pertinent suggestions for changes. To take up answer 11, due to data limitations, we unfortunately were unable to obtain data on the population with or without heavy metal exposure.

We refer to other well-designed and rigorous literature[3-5] to explore whether the risk of acquiring BV in women increases with higher heavy metal concentrations by dividing the heavy metal serum concentrations in thirds into three intervals, Q1-Q3, and calculating the ORs for Q2 and Q3 using the lowest concentration in the study population as the control group.

13. The conclusion shows that causality has been found and there are results that are not even statistically significant. In addition, the possible confounders that could exist in this analysis are not analyzed. 

Thank you for your offer and we apologize for our lack of rigorous language. In lines 329-333, we have corrected ambiguous expressions. The study is a cross-sectional study, and its results can only indicate whether heavy metals are associated with bacterial vaginitis and do not conclude whether there is a causal relationship between exposure and outcome. At the same time, we refer to several papers and include as many recognized confounders as possible as covariates. However limited to the restricted type of data in the NHANES database, we were unable to exclude all confounders. This limitation we have acknowledged inside lines 317 to 320.

14. It is necessary to determine the percentage of the population that was eligible and from which the information was obtained to analyze the data. This helps to determine that there is no over-estimation or under-estimation of the data analyzed and the results obtained. Ideally, the population analyzed should correspond to at least 80% of the eligible population. 

We appreciate your valuable feedback. Our research examined a total of 21,161 individuals from the NHANES database who had the potential to be suitable participants. Furthermore, we excluded individuals with incomplete information regarding BV (n=18,355) and serum heavy metal levels (n=99). Following the exclusion of populations lacking crucial covariate data, a total of 2,493 18-year-old to 49-year-old female participants were ultimately enrolled in our research. The population analyzed corresponds to 92% of the eligible population. Additionally, we used multiple interpolations to fill in the missing values of the covariates and used them as sensitivity analyses.

15. The graphs show group data analysis, but in this study, you collected information from each individual, so individual data analysis is essential since this is not an ecological study. Even the use of linear correlation analysis, which should be used only in the analysis of group data, is questionable. The use of correlation in these studies would introduce an ecological bias, because the relationship obtained at the group level may differ from what is determined at the individual level. 

Thanks for your careful checks. As the sample design of NHANES described each single year and any combination of consecutive years comprised a nationally representative sample of the population. However, because NHANES went to only a small number of PSUs each year, estimates for single-year data were relatively unstable (large variance estimates). In addition, releasing only 1 year of data increased the possibility of disclosure of a sample person’s identity. As a result, data were publicly released in 2-year cycles. In general, any 2-year data cycle in NHANES can be combined with adjacent 2-year data cycles to create analytic data files based on 4 or more years of data, to produce estimates with greater precision and smaller sampling error.

16. It is necessary to carry out an in-depth analysis of the biases that could interfere with the hypothesis established in this study. Mainly, in the confounding bias, because the existence of heavy metals in the serum could be only a confounder and this would invalidate the whole study. It is necessary to justify this section with criteria used in epidemiology. 

Thank you again for your positive comments and valuable suggestions to improve the quality of our manuscript. As answer 9 said, we selected these confounders based on their associations with the outcomes of interest or a change in effect estimate of more than 10%. Covariates were included as potential confounders in the final models if they changed the estimates of heavy metals on BV by more than 10% or were significantly associated with BV. Based on these two criteria, we screened for covariates that could affect the results to make our study as scientific as possible[1, 2]. Moreover, stratified analyses were generated using covariates to alleviate potential bias in the study[6]. In epidemiology, multivariable logistic regression models, a method to eliminate the effects of confounding factors, were utilized in the study to assess the correlation between heavy metals and BV, and all covariates were adjusted in model Ⅱ. We used more covariates to conduct the stratified analyses to recognize that heavy metals are not the confounders and the result of stratified analyses was stable. We modified Table 3 and described the results on lines 234 to 242. Sensitivity analysis was performed after the missing data of covariates were populated by multiple interpolations when the missing data of heavy metals and BV were eliminated.

17. Finally, it is necessary to carefully revise the entire text of the article, improve the wording, use the current bibliography, and revise the wording in English.

Thanks for your suggestions. We feel sorry for our poor writing, however, we do invite a friend of ours who is a native English speaker from the USA to help polish our article. Due to our friend’s help, the article was edited extensively. We carefully revise the entire text of the article, improve the wording, use the current bibliography, and revise the wording in English. We hope the revised manuscript will be acceptable to you.

We concur with the view that your suggestions had a beneficial effect on the quality and readability of the article. Furthermore, we consider that they enhanced the persuasiveness and veracity of the research. We would like to express our gratitude once more for your invaluable feedback. We will continue to strive for further improvements to the article's quality.

We would be delighted to hear from you if you have further suggestions or require further information. We await your feedback with interest.

Sincerely thank you!

Yu-Xue Feng

fyxgzhmu1226@outlook.com

References

1. Bjerregaard LG, Pedersen DC, Mortensen EL, Sørensen TIA, Baker JL. Breastfeeding duration in infancy and adult risks of type 2 diabetes in a high-income country. 

---

## [Decision Letter · Decision Letter 1]

6 Nov 2024

PONE-D-24-18888R1Association between heavy metal exposure and bacterial vaginosis: a cross-sectional studyPLOS ONE

Dear Dr. Guo,

Thank you for submitting your manuscript to PLOS ONE. After careful consideration, we feel that it has merit but does not fully meet PLOS ONE’s publication criteria as it currently stands. Therefore, we invite you to submit a revised version of the manuscript that addresses the points raised during the review process.

**ACADEMIC EDITOR:**

I am pleased to inform you that one reviewer already accepted the revised manuscript for publication and the other reviewer (reviewer 2) requested some revisions for future publication endorsement. Please carefully answer reviewer 2' concerns and rectify the manuscript following his/her comments.

Thank you for choosing PLOS ONE journal and best regards,

António Machado

We look forward to receiving your revised manuscript.

Kind regards,

António Machado

Academic Editor

PLOS ONE

Journal Requirements:

Additional Editor Comments:

Dear authors,

I am pleased to inform you that one reviewer already accepted the revised manuscript for publication and the other reviewer (reviewer 2) requested some revisions for future publication endorsement. Please carefully answer reviewer 2' concerns and rectify the manuscript following his/her comments.

Thank you for choosing PLOS ONE journal and best regards,

António Machado

Reviewers' comments:

Reviewer's Responses to Questions

**Comments to the Author**

1. If the authors have adequately addressed your comments raised in a previous round of review and you feel that this manuscript is now acceptable for publication, you may indicate that here to bypass the “Comments to the Author” section, enter your conflict of interest statement in the “Confidential to Editor” section, and submit your "Accept" recommendation.

Reviewer #1: All comments have been addressed

Reviewer #2: (No Response)

2. Is the manuscript technically sound, and do the data support the conclusions?

Reviewer #1: Yes

Reviewer #2: Partly

3. Has the statistical analysis been performed appropriately and rigorously? 

Reviewer #1: I Don't Know

Reviewer #2: I Don't Know

4. Have the authors made all data underlying the findings in their manuscript fully available?

Reviewer #1: Yes

Reviewer #2: Yes

5. Is the manuscript presented in an intelligible fashion and written in standard English?

Reviewer #1: Yes

Reviewer #2: Yes

6. Review Comments to the Author

Reviewer #1: can be accepted, most of the queries and issues raised have been met adequately.....................

Reviewer #2: Thank you for the corrections made, here are the details of the second revision.

When referring to “serum heavy metal exposure” it is understood as if the serum would be the one exposed. This expression is incorrect because it is the individual who has been exposed to such metals and this is reflected by elevated serum levels. But it is not the serum that is exposed (line 45).

Improve the epidemiological data in citation 3 with more information provided by an agency such as the World Health Organization or another agency that provides more recent information, such as geographic location, age, or any important clinical condition.

Indicate the consequences of disturbing the “equilibrium of human reproductive hormones” (line 89).

Search for more current references instead of 35 and 36.

Lines 320 to 324 do not account for the time constraint of the data collected. Indicate what is stated in your response in the text.

You indicate that the covariates were initially established with a literature review and international practice standards, but you do not cite them in the text. They should be placed in the introduction to emphasize the importance of the study.

Indicate in the text the reasons why the data needed to be adjusted with the appropriate references and as explained in the response to the review.

Based on the principle that OR is a measure of association between exposure and event (OR=odds that a case was exposed/odds that a control was exposed), data from persons exposed and not exposed to heavy metals and persons who did and did not develop BV are always required to calculate the OR. With the information they have they should calculate only the possible odds. Review the basis for epidemiological calculations. Other more relevant epidemiological data such as prevalences could also be calculated.

Include the percentage of the eligible population in the text.

Indicate in the text why group analyses are performed when individual data are available. How they have responded in the review.

Analyze in depth whether the statistical and epidemiological estimates provided are adequate according to the data they have and consider possible important changes.

7. PLOS authors have the option to publish the peer review history of their article (what does this mean?). If published, this will include your full peer review and any attached files.

Reviewer #1: No

Reviewer #2: **Yes: **Sandra Pamela Cangui Panchi

---

## [Author Response · Author response to Decision Letter 1]

26 Nov 2024

Responses to reviewers' comments

Dear reviewers.

We appreciate your professional review of our article It is with excitement that I resubmit to you a revised version of the manuscript (Manuscript No.: PONE-D-24-18888R1). Thank you for giving me the opportunity to revise and resubmit this manuscript. As you know, several problems need to be addressed. Your comments have greatly improved the quality of our article. 

We have carefully reviewed the comments and have revised the manuscript accordingly. I have responded specifically to each suggestion below, beginning with your own. Our responses are given in a point-by-point manner below. To make the changes easier to identify where necessary, I have numbered them. 

The specific comments from the reviewers are highlighted in red font, while my responses are in black.

To Reviewer 2

1.When referring to “serum heavy metal exposure” it is understood as if the serum would be the one exposed. This expression is incorrect because it is the individual who has been exposed to such metals and this is reflected by elevated serum levels. But it is not the serum that is exposed (line 45).

Thanks for your suggestions. We feel sorry for the improper wording. Based on your comments, we have made the corrections to harmonize the unit within the abstract. We have replaced serum heavy metal exposure with heavy metal exposure to make the presentation of the statements in the article more scientific and to avoid ambiguity (line 45).

2. Improve the epidemiological data in citation 3 with more information provided by an agency such as the World Health Organization or another agency that provides more recent information, such as geographic location, age, or any important clinical condition.

I very much appreciate the time and effort you’ve put into your comments. We agree with the reviewer’s comment concerning this issue. We improve the epidemiological data in citation 3 with more information provided by an agency such as the World Health Organization or another agency that provides more recent information, such as geographic location, age, or any important clinical condition (lines 72 to 76).

3. Indicate the consequences of disturbing the “equilibrium of human reproductive hormones” (line 89).

We acknowledge your comments very much, which are valuable in improving the quality of our manuscript. We indicated the consequences of disturbing the “equilibrium of human reproductive hormones” to make the article more readable and scientific (lines 91 to 92).

4. Search for more current references instead of 35 and 36.

Thanks for the suggestion, your suggestion means a lot to us. We searched for more current references instead of 35 and 36 (line 99).

5.Lines 320 to 324 do not account for the time constraint of the data collected. Indicate what is stated in your response in the text.

I appreciate the time and effort you’ve put into your comments. As you suggested, I have indicated what is stated in your response in the text (lines 345 to 348).

6. You indicate that the covariates were initially established with a literature review and international practice standards, but you do not cite them in the text. They should be placed in the introduction to emphasize the importance of the study.

We thank the reviewers for their constructive criticism, and time spent analyzing this manuscript. I have placed the literature review and international practice standards in the introduction to emphasize the importance of the study (lines 103 to 108).

7. Indicate in the text the reasons why the data needed to be adjusted with the appropriate references and as explained in the response to the review.

We thank the reviewer for the kind comments. We have indicated in the text the reasons why the data needed to be adjusted with the appropriate references (lines 168 to 173). Confounders refer to all the factors (both known and unknown) other than the study factors that may affect the outcome. Ensuring the authenticity and reliability of the study results is the greatest significance of controlling for confounders. Confounding was controlled by the inclusion of covariates, thereby reducing false-positive results to increase the fidelity of the study.

8. Based on the principle that OR is a measure of association between exposure and event (OR=odds that a case was exposed/odds that a control was exposed), data from persons exposed and not exposed to heavy metals and persons who did and did not develop BV are always required to calculate the OR. With the information they have they should calculate only the possible odds. Review the basis for epidemiological calculations. Other more relevant epidemiological data such as prevalences could also be calculated.

We are very grateful for your comments on the manuscript. Thanks to your suggestion and correction of our error, we reviewed the epidemiological basis, confirmed the definition of OR (correlation between exposure factors and the occurrence of event outcomes), and applied logistic regression to calculate ORs to obtain the degree of association between illness and exposure. The feasibility of this OR calculation has been demonstrated in a number of previous cross-sectional studies using data from the NHANES database. The prevalence of BV has been previously calculated using data from the NHANES database from 2001 to 2004. We have cited this study in our article (lines 74 to 76).

9. Include the percentage of the eligible population in the text.

We thank you very much for your comments for pointing out this omission. As per your suggestion, we included the percentage of the eligible population in the text (lines 138).

10. Indicate in the text why group analyses are performed when individual data are available. How they have responded in the review.

We are very grateful for your comments and thoughtful suggestions. We have already indicated in the text why group analyses are performed when individual data are available and provided an overview of the NHANES database in our article: the survey examines a nationally representative sample of about 5,000 people each year, selecting the survey sample to represent the U.S. population of all ages. Individuals are sampled to represent the group population (lines 122 to 127). 

11. Analyze in depth whether the statistical and epidemiological estimates provided are adequate according to the data they have and consider possible important changes.

Thanks for your valuable suggestions. We understand your concern about this issue. In terms of statistical methods, we used multivariate logistic regression analyses to explore the relationship between exposure factors and disease, stratified analyses to explore what was happening in different subgroups, multiple interpolation methods, and sensitivity analyses to further validate the scientific validity of the findings, and finally curve fitting to visualize the relationship between exposure and outcomes. In addition, we reviewed a large amount of literature and discussed the endocrine and immune systems to explain and support our findings more convincingly. We have described the possible important changes and limitations in the text.

I appreciate the time and detail provided by each reviewer and have incorporated the suggested changes into the manuscript to the best of my ability. The manuscript has certainly benefited from these insightful revision suggestions. I hope that the changes I’ve made resolve all your concerns about the article. I’m more than happy to make any further changes that will improve the paper and facilitate successful publication.

Sincerely thank you!

Yu-Xue Feng

fyxgzhmu1226@outlook.com

---

## [Decision Letter · Decision Letter 2]

18 Dec 2024

Association between heavy metal exposure and bacterial vaginosis: a cross-sectional study

PONE-D-24-18888R2

Dear authors,

I am pleased to inform you that the revised manuscript was accepted for publication by both reviewers.

Thank you for submitting your work to the PLOS ONE journal and best regards,

António Machado

Reviewers' comments:

Reviewer's Responses to Questions

**Comments to the Author**

1. If the authors have adequately addressed your comments raised in a previous round of review and you feel that this manuscript is now acceptable for publication, you may indicate that here to bypass the “Comments to the Author” section, enter your conflict of interest statement in the “Confidential to Editor” section, and submit your "Accept" recommendation.

Reviewer #2: All comments have been addressed

2. Is the manuscript technically sound, and do the data support the conclusions?

Reviewer #2: Yes

3. Has the statistical analysis been performed appropriately and rigorously? 

Reviewer #2: Yes

4. Have the authors made all data underlying the findings in their manuscript fully available?

Reviewer #2: Yes

5. Is the manuscript presented in an intelligible fashion and written in standard English?

Reviewer #2: Yes

6. Review Comments to the Author

Reviewer #2: (No Response)

7. PLOS authors have the option to publish the peer review history of their article (what does this mean?). If published, this will include your full peer review and any attached files.

Reviewer #2: **Yes: **Sandra Pamela Cangui Panchi

---

## [Editor Report · Acceptance letter]

29 Dec 2024

PONE-D-24-18888R2 

PLOS ONE

Dear Dr. Guo, 

I'm pleased to inform you that your manuscript has been deemed suitable for publication in PLOS ONE. Congratulations! Your manuscript is now being handed over to our production team.

Kind regards, 

on behalf of

Dr. António Machado 

Academic Editor

PLOS ONE